# Human Papillomavirus in Sinonasal Squamous Cell Carcinoma: A Systematic Review and Meta-Analysis

**DOI:** 10.3390/cancers13010045

**Published:** 2020-12-25

**Authors:** Kim J. W. Chang Sing Pang, Taha Mur, Louise Collins, Sowmya R. Rao, Daniel L. Faden

**Affiliations:** 1Department of Otolaryngology–Head and Neck Surgery, Massachusetts Eye and Ear, Boston, MA 02114, USA; kim.chang@hotmail.nl (K.J.W.C.S.P.); Louise_Collins@MEEI.HARVARD.EDU (L.C.); 2Department of Otolaryngology–Head and Neck Surgery, Boston University School of Medicine, Boston, MA 02118, USA; taha.mur@bmc.org; 3Biostatistics Center, Massachusetts General Hospital, Boston, MA 02114, USA; SRRAO@mgh.harvard.edu; 4Department of Global Health, Boston University School of Public Health, Boston, MA 02118, USA; 5Massachusetts General Hospital, Boston, MA 02118, USA; 6Harvard Medical School, Boston, MA 02115, USA

**Keywords:** human papillomavirus, sinonasal squamous cell carcinoma, prevalence, detection method, anatomic subsite

## Abstract

**Simple Summary:**

The causative role of human papillomavirus (HPV) in sinonasal squamous cell carcinoma (SNSCC) remains unclear and is hindered by small studies using variable HPV detection techniques. This meta-analysis aims to provide an updated overview of HPV prevalence in SNSCC stratified by detection method, anatomic subsite, and geographic region. From 60 eligible studies, an overall HPV prevalence was estimated at 26%. When stratified by detection method, HPV prevalence was lower when using multiple substrate testing compared to single substrate testing. Anatomic subsite HPV prevalence was higher in subsites with high exposure to secretion flow compared to low exposure subsites. HPV prevalence in SNSCC followed the global distribution of HPV+ oropharyngeal squamous cell carcinoma. Taken together, this meta-analysis further supports a role for HPV in a subset of SNSCCs.

**Abstract:**

Human papillomavirus (HPV) drives tumorigenesis in a subset of oropharyngeal squamous cell carcinomas (OPSCC) and is increasing in prevalence across the world. Mounting evidence suggests HPV is also involved in a subset of sinonasal squamous cell carcinomas (SNSCC), yet small sample sizes and variability of HPV detection techniques in existing literature hinder definitive conclusions. A systematic review was performed by searching literature through March 29th 2020 using PubMed, Embase, and Web of Science Core Collection databases. Preferred Reporting Items for Systematic Reviews and Meta-Analyses (PRISMA) guidelines were followed by two authors independently. A meta-analysis was performed using the random-effects model. Sixty studies (*n* = 1449) were eligible for statistical analysis estimating an overall HPV prevalence of 25.5% (95% CI 20.7–31.0). When stratified by HPV detection method, prevalence with multiple substrate testing (20.5%, 95% CI 14.5–28.2) was lower than with single substrate testing (31.7%, 95% CI 23.6–41.1), highest in high-exposure anatomic subsites (nasal cavity and ethmoids) (37.6%, 95% CI 26.5–50.2) vs. low-exposure (15.1%, 95% CI 7.3–28.6) and highest in high HPV+ OPSCC prevalence geographic regions (North America) (30.9%, 95% CI 21.9–41.5) vs. low (Africa) (13.1, 95% CI 6.5–24.5)). While small sample sizes and variability in data cloud firm conclusions, here, we provide a new reference point prevalence for HPV in SNSCC along with orthogonal data supporting a causative role for virally driven tumorigenesis, including that HPV is more commonly found in sinonasal subsites with increased exposure to refluxed oropharyngeal secretions and in geographic regions where HPV+ OPSCC is more prevalent.

## 1. Introduction

Human papillomavirus (HPV) has been identified as an etiological factor in a subset of head and neck squamous cell carcinomas (HNSCC). HPV-driven tumors arise predominately in the oropharynx (oropharyngeal squamous cell carcinomas (OPSCC)), but also in epithelial-derived tumors of the oral cavity, larynx and nasopharynx, albeit at significantly lower prevalence [1,2]. OPSCC driven by HPV (HPV+ OPSCC) has unique biology, epidemiology, and clinical behavior compared to OPSCC driven by carcinogen exposure. Further, and perhaps most importantly, HPV+ OPSCC has improved treatment response and overall survival [3,4,5]. At this time, detection of HPV in OPSCC is one of the only clinically utilized biomarkers in HNSCC.

The first evidence for a potential etiological role of HPV in sinonasal squamous cell carcinoma (SNSCC) tumorigenesis arose in 1983 with the detection of HPV DNA by Syrjänen et al. [6]. Since this time, mounting histologic and epidemiologic evidence suggests a subset of SNSCCs may be HPV-driven, and that similar to HPV+ OPSCC, HPV detection in SNSCC may be a biomarker for improved survival [7,8,9,10,11]. However, small sample sizes and variable HPV detection techniques, each with wide ranges in sensitivity and specificity, continue to hinder definitive conclusions. Because of this, and: (1) improvements in HPV detection techniques and (2) the changing prevalence of HPV+ OPSCC in the population, we performed a meta-analysis of HPV in SNSCC, identifying 1458 cases for inclusion. In addition to establishing a new point prevalence of HPV in SNSCC, using by far the largest cohort to date, we also test orthogonal hypotheses which would support a role for HPV-driven tumorigenesis in SNSCC, including that HPV prevalence will be highest in: (1) subsites of the sinonasal cavities with the highest exposure to refluxed secretions from the oropharynx and (2) geographic regions of the world with the highest HPV+ OPSCC prevalence.

## 2. Materials and Methods

A systematic review was performed by a medical librarian (L.C.) following the guidelines of the Preferred Reporting Items for Systematic Reviews and Meta-Analyses (PRISMA) [12].

### 2.1. Literature Search

A search of published studies in Medline via Legacy PubMed (1946-), Embase.com (1947-), and Web of Science Core Collection (1900-) was performed on 6 February 2020 to identify relevant articles. Search strategies were customized for each database (Methods S1). Each search utilized a combination of controlled vocabulary and keywords focused on the concepts human papillomavirus, squamous cell carcinoma, and sinonasal. The search was constructed to exclude non-human studies. No filters for language, study design, date of publication, or country of origin were used in the search producing 1177 articles (Figure 1). All references were exported into EndNote X7.8. Duplicates were removed first by the automated process in EndNote and then manually by the librarian leaving 730 articles, which were exported into Covidence for study screening, selection, and data extraction. The search was re-run on 29 March 2020 to update for the most recent literature rendering 14 additional articles. Three subsequent articles were found through searching the references of included articles making up a total of 747 articles for screening.

### 2.2. Study Selection

Studies examining both SNSCC and HPV status in adult patients were considered eligible for inclusion. Included SNSCC histology subtypes were non-keratinizing, keratinizing, papillary, and basaloid squamous cell carcinoma. SCC in which the histological subtype was not further specified was also considered eligible. Other SCC subtypes such as adenosquamous and multi-phenotypic sinonasal carcinoma were excluded along with studies not listing HPV detection methods or discussing cancers originating from the nasopharynx, nasal vestibule, nasal ala or skin.

Extracted data comprised geographic region of the study, histology, anatomic subsite, HPV status, HPV genotype, and HPV detection method. During the screening, any study written in a language other than English, Dutch, Arabic, or German (languages spoken by the authors) were excluded. Titles and abstracts were screened by two authors independently (K.C.S.P. and T.M.) for full text review. The same two authors independently conducted the full text review. Any disagreements in the screening process were settled by discussion and consensus between the two authors. Disagreements that could not be settled in this manner were settled in consultation with a third author (D.F.). All eligible studies were screened for duplicate data by comparing authors, timeframe of data collection, and outcomes. After full text screening, 69 studies remained for the quantitative synthesis.

### 2.3. Statistical Analysis

Comprehensive Meta-Analysis (CMA) v3 (Biostat, Englewood, NJ, USA, 2013) was used to conduct the statistical analysis. To minimize distortion of the results by outliers the CMA program excludes all studies with a sample size of one patient. Using the random-effects model, HPV prevalence estimates including 95% confidence intervals (CI) were computed from sample size and event rates. In case of an event rate of 0% or 100%, the CMA program adds 0.5 to event and non-event values for computation of logit event rates and its variance. HPV prevalence was stratified by detection method, anatomic subsite, and geographic region for descriptive comparison. Subsequently, separate meta-regressions were performed to test the association of each study characteristic with HPV prevalence estimates. Interstudy variability and between-study variance were assessed by Cochran’s Q statistic [13,14]. The percentage of variation explained by true heterogeneity opposed to sampling error was calculated with the I^2^ statistic [13]. Potential publication bias was evaluated by generating a funnel plot and assessing its asymmetry with Egger’s Test [15], Begg and Mazumdar Rank Correlation Test [16], and Duval and Tweedie’s “Trim and Fill” method [17]. A sensitivity analysis was performed by removing one study at a time to assess the influence of each individual study on the combined HPV prevalence. We assume a two-sided *p* < 0.05 to be significant.

## 3. Results

### 3.1. HPV Prevalence

A total of 69 studies were included in the meta-analysis containing a total of 1458 patients with SNSCC (Table 1). There were 324/1458 HPV-positive cases comprising a crude HPV prevalence of 22.2%. After removal of all studies with a sample size of one patient, 60 studies remained for statistical analysis. Estimated HPV prevalence rates ranged from 5.0 to 94.4%. Using the random-effects model, an overall prevalence rate was estimated at 25.5% (95% CI 20.7–31.0) (Table 1).



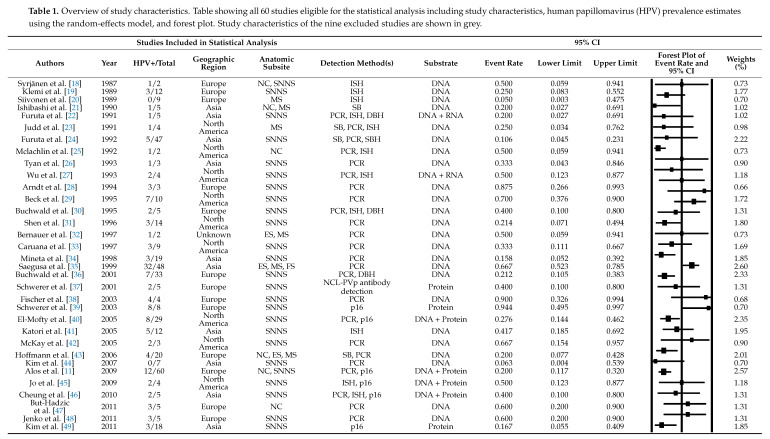





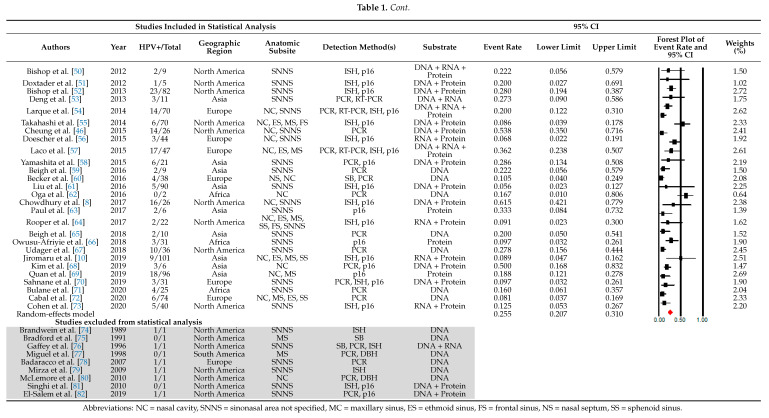



### 3.2. HPV Detection Method

There were 32 DNA-based studies, six protein-based studies, 12 DNA + protein-based studies, two DNA + RNA-based studies, four RNA + protein-based studies, and two DNA + RNA + protein-based studies. There were no studies using only RNA-based detection methods. DNA-based detection methods included DNA in situ hybridization (ISH), Southern blotting (SB), polymerase chain reaction (PCR), dot-blot-hybridization (DBH), and slot-blot-hybridization (SBH). RNA-based detection methods included RNA ISH and reverse transcriptase PCR (RT-PCR). Protein-based detection methods included p16 IHC and NCL-PVp antibody detection. Due to the small sample size in the majority of the subgroups, we categorized studies as either detecting a single HPV substrate (DNA-based or protein-based) or detecting a combination of HPV substrates (DNA + protein-, DNA + RNA-, RNA + protein-, or DNA + RNA + protein-based). Two studies, Deng et al., (2013) [53] and Larque et al., (2014) [54], first assessed HPV-positivity using DNA-based testing and only conducted additional RNA-based testing on the HPV-positive tumors. Since these studies show selection bias, both were excluded from this analysis. Additionally, when looking at the distributions of other variables in the two subgroups, only the single-agent testing group contained studies conducted in Africa. Since the three African studies reported a low HPV prevalence and thereby bias the results, they were also excluded from this analysis. The results of the 55 remaining studies are shown in Table 2.

As expected, HPV prevalence was lower when a combination of detection methods was used, reflecting fewer false positives, compared to when DNA- or protein-detection was used alone (Table 2A)—this difference was of border-line significance (Q = 3.88, *p* = 0.05, I^2^ = 70.5%). Since the gold standard for determining a tumor is HPV-driven in OPSCC is RNA-based testing, we also wanted to compare RNA-based to DNA- and protein-based detection techniques. This is of particular importance as the diagnostic utility of p16 overexpression in SNSCC remains unclear. As there were no single RNA-testing studies, we split the multi-agent testing group into either RNA (RNA + DNA, RNA + protein, DNA + RNA + protein) or no RNA (DNA + protein). Again, in line with our expectations, the RNA group yielded the lowest HPV prevalence (Table 2B); however, the difference across the three groups was not statistically significant (Q = 4.60, *p* = 0.10, I^2^ = 70.5%).

### 3.3. Anatomic Subsite

We next considered HPV prevalence stratified by sinonasal subsite. These data existed in 20 studies. The remaining 40 studies did not specify sinonasal subsite. We categorized anatomic subsites as either high-exposure to refluxed oropharyngeal secretion flow (nasal cavity and ethmoids), or low-exposure (maxillary, frontal, and sphenoid sinuses). In line with our hypothesis, analysis using the random-effects model yielded the highest HPV prevalence in high-exposure subsites (37.6%, 95% CI 26.5–50.2) and lower prevalence in less exposed subsites (15.1%, 95% CI 7.3–28.6) (Figure 2A) with the prevalence of unspecified sinonasal area (likely a combination of all subsites) in the middle (25.6%, 95% CI 20.1–31.7) (Figure 2B).

### 3.4. Geographic Region

Data for HPV prevalence stratified by geographic region were available for 59 studies. Three studies were conducted in Africa, 18 in North America, 19 in Asia, and 19 in Europe. No studies were available for analysis from South America or Oceania. Using the random-effects model, the highest HPV prevalence estimate was found in North America (30.9%, 95% CI 21.9–41.5), in line with existing literature using the National Cancer Database (32.0%) [9,83], and the lowest in Africa (13.1%, 95% CI 6.5–24.5). These trends mirrored HPV prevalence of OPSCC after matching for countries of origin (Figure 3). Remarkably, when examining data from North American studies only, high risk subsites showed HPV detection rates approaching those seen in OPSCC (Figure 4B).

### 3.5. Analysis of Validity, Sensitivity, Data Trends and Publication Bias

The included studies show a significant amount of interstudy variability: Cochrane’s Q = 188.23 (*p* < 0.001); I^2^ = 68.7%. Studies included in the subgroup analysis for detection method (single- vs. multi-agent testing) and anatomic subsite were significantly heterogeneous (Q = 3.88, *p* = 0.05; I^2^ = 70.5% and Q = 6.81, *p* = 0.03, I^2^ = 63.5%, respectively), but not for geographic regions (Q = 5.82, *p* = 0.12). Meta-regression results indicated that both detection method ((Q = 3.54, *p* = 0.06), with a border-line significance, and anatomic subsite (Q = 6.33, *p* = 0.04) were associated with the outcome (Appendix A). However, due to limited number of studies and sample sizes we were unable to include all three covariates in one model. Our outcomes show a constant presence of interstudy variability limiting the conclusions which can be drawn. Of note, intra-study sample size increased across time, as did the use of RNA and multi-substrate testing, leading to more high yield studies (Appendix A).

Sensitivity analysis conducted by removing one study at a time showed a relatively stable HPV prevalence estimate with the random-effects model with the lowest HPV prevalence of 24.3% (95% CI 19.9–29.3) when the study by Saegusa et al., (1999) [35] was removed and the highest HPV prevalence of 26.2% (95% CI 21.3–31.7) when the study of Liu et al., (2016) [61] was removed.

Begg and Mazumdar’s rank correlation test yielded a Kendall’s tau b with continuity correction of 0.245 with a one-tailed *p*-value of 0.003 showing significant funnel plot asymmetry (Appendix A). Duval and Tweedie’s Trim and Fill method using the random-effects model imputed 14 missing studies and yielded an adjusted HPV prevalence of 19.1% (95% CI 14.8–24.3). However, Egger’s test showed no statistical evidence for publication bias with an intercept (B0) of 46.3 (95% CI −0.540–1.465) with t = 0.924, df = 58, and one-tailed *p*-value of 0.180.

## 4. Discussion

HPV+ OPSCC is increasing in prevalence across the world and has now surpassed cervical cancer as the most common HPV-mediated malignancy. HPV status is a critical biomarker for OPSCC, signifying improved response rates to treatment and improved survival [3]. HPV+ OPSCC now necessitates its own staging criteria in the American Joint Committee on Cancer, Eighth edition, separate from the OPSCC caused by carcinogen exposure, and the rest of HNSCCs [87]. Because of the strong prognostic implications of HPV compared to carcinogen-driven tumorigenesis in HNSCC, considerable interest exists in the role of HPV in subsites outside the oropharynx. Numerous distinct cohorts of patients with HNSCC who lack carcinogen exposure have been interrogated as potential HPV-mediated tumors, for example, oral tongue squamous cell carcinoma in young non-smoking patients. However, multiple studies have refuted this hypothesis [88,89,90]. Overall, less than 5% of HNSCCs outside the oropharynx appear to be HPV-driven, based on genomic interrogation of over 500 HNSCCs in The Cancer Genomes Atlas (TCGA) [91]. Of importance, the TCGA cohort excluded rare subsites, including SNSCC.

Cancer of the nasal and paranasal sinuses account for <3% of head and neck tumors, with SNSCC being the most common histologic subtype, comprising about half of cases [9,92,93,94]. Due to the nonspecific nature of initial symptoms, patients often present at a locally advanced stage [9,95,96]. The proximity of these malignancies to critical anatomic structures means treatment carries significant morbidity and poor overall survival [9,10,95,96,97]. Interest in the role of HPV in SNSCC spans back numerous decades [98]. A wide range of HPV detection rates in SNSCC have been reported, varying from 0 to 100% [8,10,20,38,44,54,67,69,99]. Major barriers to progress in defining the role of HPV in SNSCC include: (1) the relative rarity of SNSCC and thus published literature often utilizing small, single institution cohorts, (2) the use of disparate HPV detection techniques with significant variation in sensitivity and specificity for HPV, (3) few studies using “gold standard” platforms (E6/E7 mRNA detection with ISH or RT PCR) to demonstrate transcriptionally active HPV, ruling out a contamination or “bystander” infection and (4) exclusion of SNSCC from TCGA and a dearth of comprehensive genomic studies examining SNSCC at the DNA and RNA level [86,100,101,102].

Our systematic literature review revealed only one meta-analysis, using < 500 pooled cases, from 35 studies published prior to 2012 [103]. In this study, the authors calculated an overall HPV prevalence rate of 27.0%. In addition to the small sample size, a number of critical limitations exist in applying the findings to our primary endpoints here, including: (1) a focus only on SNSCC arising from papillomas, which are a distinct subgroup of SNSCC and (2) a complete lack of RNA-based or multiple substrate testing studies, which are significantly more likely to approximate a “true” HPV-mediated cancer prevalence rate. Since 2012, significant interest in the role of HPV in SNSCC has led to a notable increase in the available pooled cohort for analysis (69 studies identified out of 747 screened, yielding 1458 cases). Additionally, the number of RNA-based and multi-substrate testing studies, and size of cohorts published have both increased across time, yielding more high quality studies.

Here, we aimed to provide an updated overall point prevalence for HPV detection in SNSCC using a larger and more contemporary cohort, and an estimate of the prevalence of SNSCCs likely to be driven by HPV, using multi-substrate testing as a benchmark, to increase specificity above DNA detection alone. Additionally, we hypothesized that if a subset of SNSCCs is indeed driven by HPV, orthogonal data should support this, including: (1) increased HPV prevalence in sinuses with more exposure to refluxed secretion flow from the oropharynx and (2) higher HPV+ SNSCC prevalence rates in regions of the world with higher HPV+ OPSCC prevalence. Using the random-effects model, we found an overall HPV point prevalence of 26%. When sub-stratified by single- vs. multi-substrate testing, we identified a prevalence of 21% for multi-substrate testing, which we posit should represent a more accurate number for estimating HPV-driven SNSCC from the cohort available here. As expected, prevalence decreased in a stepwise fashion when tests with increasing specificity were applied. Unfortunately, there were no studies using gold-standard RNA-based detection techniques alone, which met inclusion criteria for the study. This highlights the need for additional, large cohort studies using RNA-based detection methods. It should further be noted that while p16 overexpression is a widely recognized surrogate marker for high risk HPV in OPSCC, whether p16 is a sensitive and specific marker for SNSCC has not been well established. In our systematic literature review we found 18 studies using both p16 IHC and DNA- and/or RNA-based HPV testing. However, the majority of the studies were too small to make a statement on the reliability of p16 IHC. Eight studies reported correlations of 69–100% between p16 overexpression and positive HPV status [10,11,19,22,81,84,90,97]. Reported sensitivity for p16 ranged from 88 to 100% and specificity from 67 to 100% [10,11,19,84,97]. Predictive values were calculated in three studies, all comparing p16 IHC to RNA-based HPV testing [10,84,90]. Positive predictive values ranged from 50 to 94% and negative predictive values ranged from 94 to 100%.

In line with our hypotheses, we found the highest HPV prevalence in sinonasal subsites with the greatest exposure to refluxed secretion flow from the oropharynx and the lowest prevalence in sinuses more remote from routine exposure, i.e., the more anterior, cranial and posterior sinuses, each of which also possess restrictive ostium. High-exposure subsites had HPV prevalence rates more than double low-exposure sites (38% vs. 15%). Interestingly, HPV prevalence rates by subsite mirror reported overall survival rates stratified by subsite with frontal and sphenoid sinuses having the lowest survival and nasal cavity having the highest survival [104,105]. Considering HPV+ OPSCC’s improved survival compared to non-HPV OPSCC, in part due to improved responsiveness to current treatment schemas and existing studies suggesting HPV+ SNSCCs have improved survival compared to non-HPV SNSCC, additional studies will be needed to parse out the relationship between sinonasal subsite, HPV status and survival [9].

Additionally, we found considerable variation by geographic region, which aligned with HPV+ OPSCCs rates. For example, overall HPV prevalence was highest in North American studies and lowest in African studies in both OPSCC (using previously published cohorts) and SNSCC, with SNSCC HPV prevalence rates in both cohorts being approximately 50% of the OPSCC rate. Remarkably, when restricting to examination of high risk subsites in North American studies (those most likely to be HPV positive), prevalence rates approximate HPV prevalence rates in the oropharynx in some parts of the US (52%). Additional large cohort studies using RNA-based detection techniques are needed to evaluate if these findings remain true, as sample sizes available for these sub-analyses are small.

This study has a number of limitations which relate to the status of exiting literature. First, sample sizes of available studies are small, with 27 of the 60 studies included in the analysis having a sample size of under ten patients (36 of 69 studies, total). Second, there is significant heterogeneity of HPV testing methodologies, each of which have variable sensitivity and specificity. Additionally, the majority of studies (37/69) use DNA testing alone, which may not represent a tumor driven by HPV but instead contamination or a bystander infection. Small sample sizes and heterogeneity of the datasets, as highlighted by the Q and I^2^ statistics make definitive conclusions challenging (Appendix A). Despite this, findings of this analysis are in line with our pre-existing hypotheses, increasing confidence in our conclusions. Due to the high levels of heterogeneity between datasets and the large number of missing variables needed to accurately perform subgroup analyses, we chose not to evaluate certain factors which are likely to impact true HPV+ SNSCC prevalence rates such as association with papilloma, histologic subtypes and viral genotype [7]. Of note, recent reports have highlighted SNSCCs which arise from inverted papillomas and are associated with low risk HPV types 6/11 [106]. The role, and prevalence, of low risk HPVs in SNSCCs were not evaluated here. Future studies should focus on reporting the results of genotype-specific assays. Lastly, a recently recognized histologic variant of sinonasal cancer is HPV-related multiphenotypic sinonasal carcinoma (HMSC). HMSC is formerly known as HPV-related carcinoma with adenoid cystic-like features and is strongly associated with HPV-33 [107]. HMSC is characterized by mixed phenotypes including squamous differentiation, resembling SNSCC in some cases. While we excluded HMSC from our search, it is possible that our dataset includes HMSC mistaken for SNSCC, particularly in the studies published prior to HMSC’s first description in 2012 [108].

## 5. Conclusions

Here, we provide a new reference point prevalence for HPV in SNSCC, stratified by detection method, along with orthogonal data supporting a causative role for virally driven tumorigenesis in SNSCC. Small sample sizes, high interstudy variability and missing data such as genotype-specific incidence highlight the need for large prospective evaluations of HPV in SNSCC and detailed genomic studies to further clarify the role of HPV in SNSCC.

## Figures and Tables

**Figure 1 cancers-13-00045-f001:**
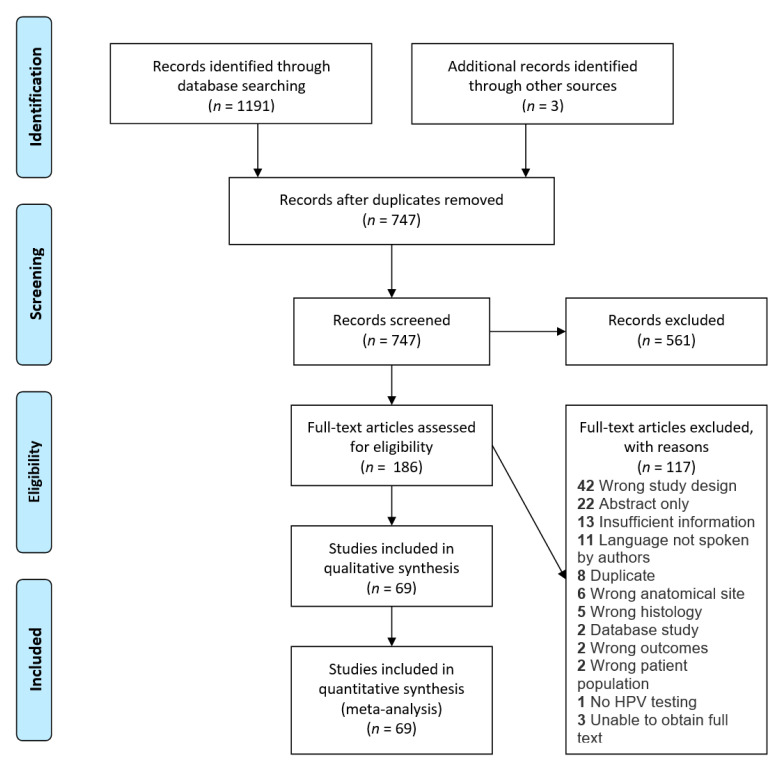
Preferred Reporting Items for Systematic Reviews and Meta-Analyses (PRISMA) flowchart depicting the study selection process.

**Figure 2 cancers-13-00045-f002:**
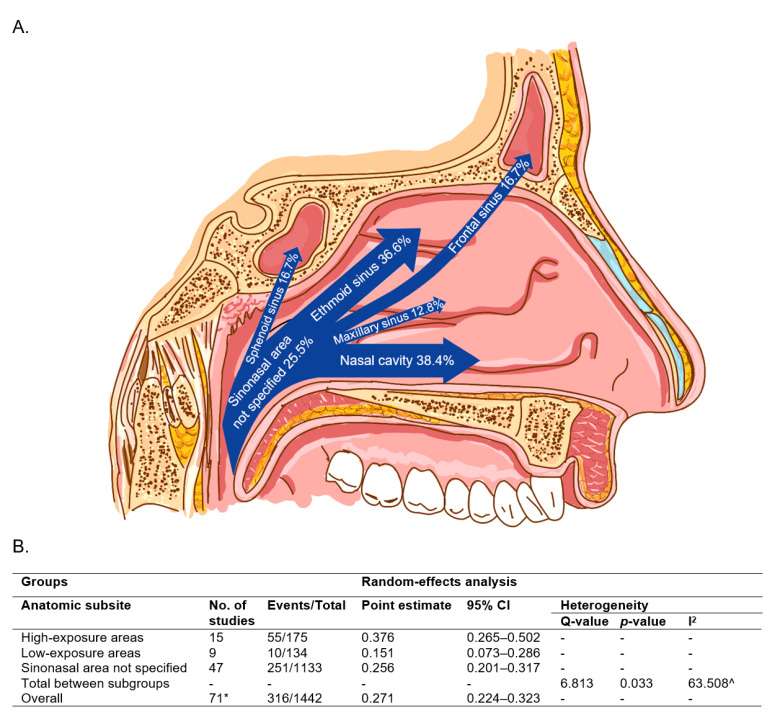
HPV prevalence distribution by sinonasal anatomic subsite. (**A**). Sagittal section of the nasal cavity with arrows displaying the entry and distribution of HPV prevalence estimates by anatomic subsite using the random-effects model. (**B**). HPV prevalence estimates stratified by high- and low-exposure anatomic subsites. * With multiple subgroups in one study, Comprehensive Meta-Analysis (CMA) program will see each subgroup as a separate study. Hence, a total of 71 studies here. ^ Only calculated using the fixed-effects model.

**Figure 3 cancers-13-00045-f003:**
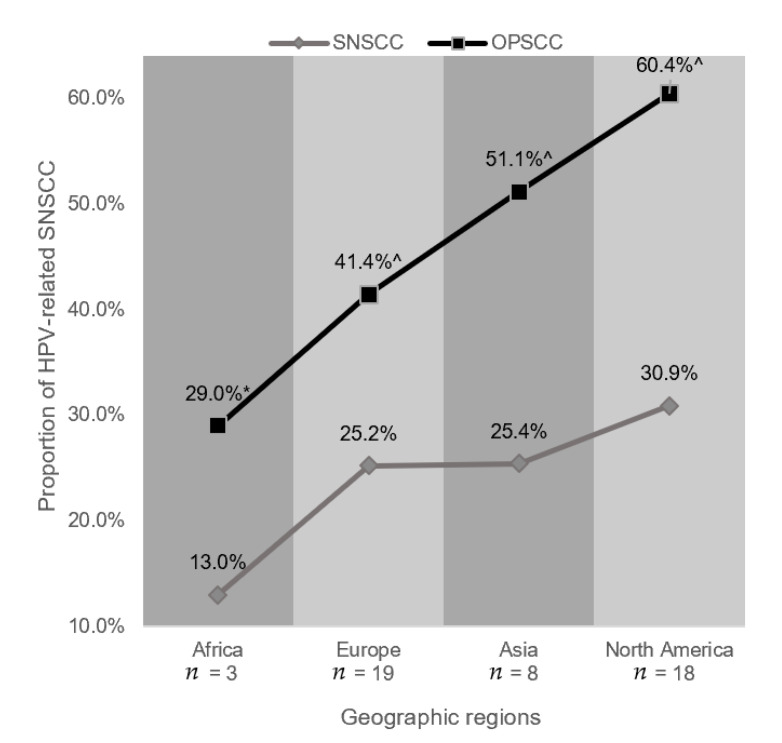
HPV prevalence in sinonasal squamous cell carcinoma (SNSCC) stratified by geographic region for oropharyngeal squamous cell carcinoma (OPSCC) and SNSCC demonstrating paired prevalence rates using the random-effects model. Studies from India and Japan were removed from the Asian SNSCC data as they were felt to introduce bias as these counties were not represented in the OPSCC data. * Source: Jalouli et al., (2010) [84] and Jalouli et al., (2012) [85]. ^ Source: Ndiaye et al., (2014) [86].

**Figure 4 cancers-13-00045-f004:**
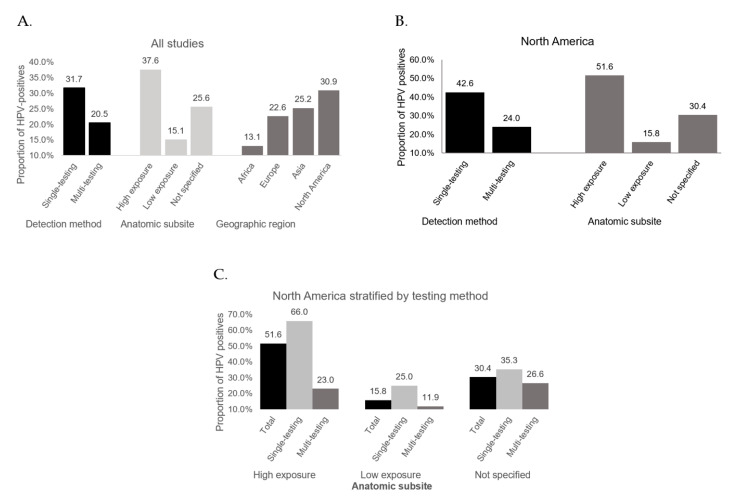
HPV prevalence by subgroup. (**A**). Bar chart depicting an overview of HPV prevalence per subgroup. (**B**). HPV prevalence per subgroup using only data from North America. (**C**). North American data stratified by testing method.

**Table 2 cancers-13-00045-t002:** HPV Prevalence estimates stratified by detection method. (**A**). HPV prevalence stratified by single-agent testing and multi-agent testing. (**B**). HPV prevalence stratified by single-agent testing, multi-agent testing using RNA, and all multi-agent testing not using RNA.

Groups				Random-Effects Analysis		Heterogeneity		
Detection Method		No. of Studies	Events/Total	Point Estimate	95% CI	Q-Value	*p*-Value	I^2^
A	Single testing	35	152/583	0.317	0.236–0.411	-	-	-
Multi-testing		20	142/727	0.205	0.145–0.282	-	-	-
Total between study		-	-	-	-	3.878	0.049	-
B	Single testing	35	152/583	0.317	0.236–0.411	-	-	-
Multi-testing without RNA		12	101/455	0.233	0.150–0.343	-	-	-
Multi-testing with RNA		8	41/272	0.165	0.088–0.287	-	-	-
Total between study		-	-	-	-	4.595	0.101	-
Overall		55	294/1310	0.261	0.208–0.322	-	-	70.452 ^

^ Only calculated using the fixed-effects model.

## Data Availability

All data generated or analyzed during this study are included in this published article (and its Appendix A).

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
