# Peer review of "Human Papillomavirus in Sinonasal Squamous Cell Carcinoma: A Systematic Review and Meta-Analysis"

_cancers, 2020, doi:10.3390/cancers13010045_

Round 1
Reviewer 1 Report
Overview
The authors performed a systematic review and meta-analysis to provide overall prevalence for human papilloma (HPV) virus in sinonasal squamous cell carcinoma (SNSCC) in terms of detection method, anatomic subsite and geographic lesions. The information obtained from this study would be useful if the method of the study is appropriate from the view point of clinical head and neck oncology.
Major Points
#1. The authors picked up 69 studies reporting HPV status in SNSCC. The authors in some reports among them defined only p16 protein overexpression as HPV related SNSCC. It is well known by head and neck oncologist that p16 protein can be a surrogate marker in oropharyngeal carcinoma, after a bunch of studies concerning about it and a lot of discussion made by investigators. Is there any solid evidence demonstrating that p16 overexpression can be a surrogate marker of HPV related SCC in nasal paranasal cavity?
#2. When we study about HPV, the information about genotype of HPV is thought to be indispensable. As we know HPV virus has more than 100 of subtypes and exists everywhere in our circumstances. Based on difference in ability to infect mucosal surface and DNA sequence, HPV virus can be classified into low risk and high risk.
Low risk type HPV, such as HPV6 and HPV11 cause benign neoplasms, such as papilloma. On the other hand, high risk HPV, such as HPV16 and HPV18 cause malignant tumor, squamous cell carcinoma in oropharynx. In sinonasal cavity, papilloma is thought to be a common disease, some of which may be caused by low risk HPV and 10-30% of papilloma generates squamous cell carcinoma with the lapse of time. On the other hand, high risk HPV could generate de novo SCC if it infects in the mucosa of nasal paranasal cavity. There seems to be 2 different kinds of carcinogenesis in nasal paranasal cavity.
I really want to know about which genotype is dominant in nasal paranasal carcinogenesis and I do not trust a report that ignored the information about HPV genotype.
Minor Points
P11 L312: Conclusion is missing
Reviewer 2 Report
The authors give an overview of HPV prevalence in SNSCC by meta-analyzing 60 eligible studies . They conclude from the data that HPV association is relevant for SCSCC, in particular in distinct subsites and geographic regions. The manuscript is well-written and the results are concisely presented. However, in some parts the discussion lacks specific conclusions drawn from the statistical results.
- The authors correlate HPV prevalence rates in distinct anatomic subsites with OS (ll. 278/279). However, they do not refer to current treatment schemes. The more favorable outcome of OPSCC is not least due to increased sensitivity to standard treatment, especially irradiation, compared to HNSCC not associated with HPV. The authors should comment on this well-accepted correlation. Is there any information about treatment regimens in the studies. If yes, is there any association between HPV, anatomic subsite and therapeutic scheme/ outcome/ prognosis that can be statistically assessed.
- How is the increased ratio of HPV-driven SNSCC in the nasal cavity and ethmoids being explained (l. 274)? Is the virus supposed to be transmitted by the air? If so, is there any evidence for this assumption in the literature?
- Why are RNA-based detection techniques considered as the gold standard (l. 160)? RNA expression of the E6/E7 oncogenes generally is considered to be the gold standard for detection of oncogenic HPV infections on fresh-frozen samples. Was the sample fixation (snap-frozen or FFPE) taken into account? The authors should also comment on the relevance of p16INK4a immunohistochemistry as a surrogate marker in HPV-associated SNSCC.
- What is exactly meant by secretion flow? The authors should specify the term and clarify why they consider subsites with secretion flow (l. 169), i.e. nasal cavity and ethmoids as high risk. If exposure with paranasal and nasal secretion is assumed to trigger carcinogenesis due to HPV load at these high risk subsites the source of HPV might as well be maxillary and frontal sinuses as those drain via Meatus nasi medius along with the frontal ethmoid sinus.
Reviewer 3 Report
This manuscript by K. Pang et al is a meta-analysis of 60 studies that estimates the prevalence of HPV in sinonasal squamous cell carcinoma (SNSCC). An overall HPV prevalence in SNSCC is estimated to be 26%. Comparisons in prevalence measurement are made based on detection method(s) used, high vs. low airflow sinonasal subsites, and geographic region. Unsurprisingly, prevalence was lower with RNA-based detection methods, at lower airflow sinonasal subsites, and in world regions with lower prevalence of HPV+ oropharynx cancer. The data appears not to have been sufficient to draw conclusions regarding outcomes of HPV+ vs. HPV- SNSCCs. The authors conclude that the study “provides a new reference point prevalence for HPV in SNSCC along with orthogonal [subsite and geographic] data supporting a causative role for virally-driven tumorigenesis.” Strengths include a sophisticated and well-executed meta-analysis design addressing a relevant question related to HPV-related tumorigenesis in the head and neck. A weakness is the fairly modest new insight provided into potential causality of HPV in SNSCCs. Some specific points of critique are as follows:
- The calculated 26% HPV prevalence highlighted by the meta-analysis does not imply causality, and thus this finding seems to over-emphasized throughout the manuscript. The studies that included RNA-based detection support a viral etiology for a smaller subset of tumors (16.5%), and that number is probably the more informative piece of data to highlight.
- It is unclear to the reviewer how the subsite and geographic prevalence comparisons provide much support for a causal role of HPV in SNSCC. Even if high risk HPV is a mere passenger phenomenon in most or all SNSCCs where it is detectable, one would expect to see the same subsite and geographic differences.
- Studies using “protein-based” detection methods appear to be defined as those with positive p16 IHC or NCL-PVp antibody detection. However, the significance of those two methods is rather disparate, and so that grouping does not make much sense to the reviewer. In absence of viral etiology being predominant at an anatomic site, positive p16 IHC is very likely to be false positive as a surrogate biomarker. By contrast, whereas antibody detection of viral particles arising from tumor cells is rather strongly suggestive of viral etiology.
- In discussing the geographic data, the authors state that “when examining data from North American studies only, high risk subsites showed HPV prevalence rates approaching those seen in OPSCC (Figure 4).” This appears to be an overstatement and is potentially misleading, given the relatively scant evidence for viral etiology in SNSCC relative to that available for OPSCC.
- It is notable that multiphenotypic sinonasal cancers, which now have a fairly well supported HPV-related etiology, were excluded from the study. Some of these cancers have a large component of squamous differentiation and thus may exist as a spectrum with the HPV-related purely squamous malignancies analyzed here. This might make a worthwhile discussion point.
Round 2
Reviewer 1 Report
#1.Thank you to the reviewer for this thoughtful comment. As noted, p16 positivity is a widely accepted and recommend marker for HPV driven tumorigenesis in the oropharynx. It is the recommended diagnostic test by the American Association of Pathologists. Because of this, in the SNSCC, p16 is the most commonly investigated marker. Regardless of anatomic site in the upper aerodigestive tract, HPV induces p16 overexpression. Thus, we expect the rate of true negatives and true positives to be similar to the oropharynx while the sensitivity and positive predictive value will be lower due to the lower incidence of true positives. Whether p16 can be a surrogate marker for sinonasal tumors has not been well-established in the literature. In our systematic literature review we found 18 studies using both p16 IHC and DNA- and/or RNA-based HPV testing. However, the majority of the studies were too small to make a statement on the reliability of p16 IHC. Eight studies reported correlations of 69-100% between p16 overexpression and positive HPV-status [10,11,19,22,81,84,90,97]. Reported sensitivity for p16 ranged from 88-100% and specificity from 67-100% [10,11,19,84,97]. Predictive values were calculated in three studies, all comparing p16 IHC to RNA-based HPV testing [10,84,90]. Positive predictive values ranged from 50-94% and negative predictive values ranged from 94-100%. Thus, for now, we are limited to the available literature. To overcome this issue we explicitly analyzed our data using a number of different grouping based on substrate tested. This included: 1) single vs. multiple testing and 2) single vs. multiple testing without RNA and multiple testing with RNA. All of this data is openly presented in the manuscript for the reader along with a discussion of why we chose to group the data into so many different analyses. As this is a meta-analysis, we are limited by the available data. We have done our best to present the data in an objective way and provide our interpretation of the data. We have added the line “This is of particular importance as the diagnostic utility of p16 overexpression in SNSCC remains unclear.”, to further emphasize the issue of using p16 status as a diagnostic test, in the results section and added the following to the discussion as well: “ It should further be notes that while p16 overexpression is a widely recognized surrogate marker for high risk HPV in OPSCC whether p16 can be a surrogate marker for sinonasal tumors has not been well-established. In our systematic literature review we found 18 studies using both p16 IHC and DNA- and/or RNA-based HPV testing. However, the majority of the studies was too small to make a statement on the reliability of p16 IHC. Eight studies reported significant correlations of 69-100% between p16 overexpression and positive HPV-status [10,11,19,22,81,84,90,97]. Reported sensitivity for p16 ranged from 88-100% and specificity from 67-100% [10,11,19,84,97]. Predictive values were calculated in three studies, all comparing p16 IHC to RNA-based HPV testing [10,84,90]. Positive predictive values ranged from 50-94% and negative predictive values ranged from 94-100%”.
It was great that the authors demonstrated the sensitivity, specificity, PPV and NPV of p16 protein expression in detecting HPV infection by referring several past reports.
This is what I would like the authors to clarify in this manuscript.
I appreciate the authors’ response to my comment at this point.
#2.While we agree that genotype is an important piece of information, as this is a meta-analysis, we are again limited by the available data. We collected and analyzed this data as part of the analysis. What we determined is that the information available for genotype was too limited, or not able to be interpreted in a way that allowed it to be included in the meta-analysis. Specifically, 48/69 studies reported HPV genotype in some fashion. However, the vast majority did not include information to a genotype level. For example, they would report their assay was positive for a sample for “all-high risk genotypes” or “positive for 16/18/33/35” making it not possible to report genotype specific data. Thus, we chose not to report this data as it was not interpretable. We have added the following to speak to this point in the discussion: “We also chose not to focus on HPV genotypes again due to the large number of studies missing this information. Future studies should focus on reporting the results of genotype specific assays. “
When I read the authors response, I realized that the majority of HPV testing in accumulated reports seemed to target high risk HPV only.
As I mentioned in the previous review, we sometimes experienced nasal squamous cell carcinoma arising from inverted papilloma (IP) and IP is causable by low risk HPV, such as HPV6/11. Some investigators detected HPV6/11 in SCC arising from IP in a certain population. Readers may wonder if the prevalence of HPV infection might be underestimated when they targeted high risk HPV only. On the other hand, some investigators reported that low risk HPV was associated with IP only and high risk HPV infection may play a certain role in transformation from benign lesion to squamous cell carcinoma.
I requested the authors to make some discussion about this issue.
Author Response
Thank you to the reviewer for their comments.
- I am not sure if the reply: "It was great that the authors demonstrated the sensitivity, specificity, PPV and NPV of p16 protein expression in detecting HPV infection by referring several past reports.This is what I would like the authors to clarify in this manuscript. I appreciate the authors’ response to my comment at this point." is asking for further clarification or not. I do not believe so but please let me know if so.
-
"When I read the authors response, I realized that the majority of HPV testing in accumulated reports seemed to target high risk HPV only. As I mentioned in the previous review, we sometimes experienced nasal squamous cell carcinoma arising from inverted papilloma (IP) and IP is causable by low risk HPV, such as HPV6/11. Some investigators detected HPV6/11 in SCC arising from IP in a certain population. Readers may wonder if the prevalence of HPV infection might be underestimated when they targeted high risk HPV only. On the other hand, some investigators reported that low risk HPV was associated with IP only and high risk HPV infection may play a certain role in transformation from benign lesion to squamous cell carcinoma. I requested the authors to make some discussion about this issue."
Response: We agree that this is a very interesting topic, and have been following this literature. For example, Mehrad et al, 2020. Most of the studies we have reviewed here examine HR HPV, as you mentioned, which is by far, the most common culprit in head and neck malignancies. While we agree that there may be a rare subset of SNSCCs that are transformed IPs, and are driven by HPV 6/11, these are likely exceptionally rare. More generally speaking, this general concept remains unproven. None-the-less, we have added a brief discussion to the manuscript
Reviewer 3 Report
The issues from the initial reviewer critique have been adequately addressed.
Author Response
Thank you